# Nonstructural Protein 2 Is Critical to Infection Efficiency of Highly Pathogenic Porcine Reproductive and Respiratory Syndrome Virus on PAMs and Influence Virulence In Vivo

**DOI:** 10.3390/v14122613

**Published:** 2022-11-23

**Authors:** Jiazeng Chen, Lingxue Yu, Yanjun Zhou, Shen Yang, Yun Bai, Qian Wang, Jinmei Peng, Tongqing An, Fei Gao, Liwei Li, Chao Ye, Changlong Liu, Guangzhi Tong, Xuehui Cai, Zhijun Tian, Yifeng Jiang

**Affiliations:** 1State Key Laboratory of Veterinary Biotechnology, Harbin Veterinary Research Institute, Chinese Academy of Agricultural Sciences, Harbin 150001, China; 2Shanghai Veterinary Research Institute, Chinese Academy of Agricultural Sciences, Shanghai 200241, China; 3Jiangsu Co-Innovation Center for the Prevention and Control of Important Animal Infectious Diseases and Zoonoses, Yangzhou University, Yangzhou 225009, China

**Keywords:** PRRSV, infection efficiency, PAM, virulence

## Abstract

Porcine reproductive and respiratory syndrome (PRRS) is an important viral disease, causing significant economic losses to the swine industry worldwide. Atypical cases caused by highly pathogenic PRRS virus (HP-PRRSV) emerged in 2006 in China. The vaccine strain HuN4-F112 has been developed from the wild-type HP-PRRSV HuN4 through repeated passages on MARC-145 cells. However, the mechanisms of attenuation have yet to be defined. Previous studies have shown that the vaccine strain HuN4-F112 could not effectively replicate in porcine alveolar macrophages (PAMs). In the present study, a series of chimeric and mutant PRRSVs were constructed to investigate regions associated with the virus attenuation. Firstly, the corresponding genome regions (ORF1a, ORF1b and ORFs 2-7) were exchanged between two infectious clones of HuN4 and HuN4-F112, and then the influence of small regions in ORF1a and ORF2-7 was evaluated, then influence of specific amino acids on NSP2 was tested. NSP2 was determined to be the key gene that regulated infection efficiency on PAMs, and amino acids at 893 and 979 of NSP2 were the key amino acids. The results of in vivo study indicated that NSP2 was not only important for infection efficiency in vitro, but also influenced the virulence, which was indicated by the results of survival rate, temperature, viremia, lung score and tissue score.

## 1. Introduction

Porcine reproductive and respiratory syndrome (PRRS) is an economically devastating viral disease that has caused significant loss for the swine industry worldwide [1]. The causative agent of this disease, PRRS virus (PRRSV), was first recognized in the early 1990s in both Europe and North America [2,3,4]. PRRSV is an enveloped positive stranded RNA virus that belongs to the family *Arteriviridae*, genus *Betaarterivirus*. Characterized by its genetic and antigenic heterogeneity, PRRSV can be divided into two major genotypes: genotype 1 (Europe) and genotype 2 (North American). These two genotypes typically share ~60% nucleotide similarity over the entire genomes and can be distinguished serologically [5,6,7,8].

The PRRSV genome is approximately 15 kb in length and consists of at least 10 open reading frames (ORFs) including ORF1a, ORF1b, ORF2, ORF2a, ORF3, ORF4, ORF5, ORF5a, ORF6 and ORF7 [9,10]. ORF1a and ORF1b are replicase-associated genes that encode the polyproteins pp1a and pp1ab, respectively [11,12]. The pp1a was proteolytically processed into nine nonstructural proteins (NSPs), including three important virus proteases NSP1 (papain-like cysteine protease), NSP2 (chymotrypsin-like cysteine protease) and NSP4 (3C-like serine protease), and proteolytic cleavage of pp1ab results in the following products: NSP9 (viral RNA-dependent RNA polymerase, RdRP), NSP10 (RNA helicase), NSP11 (endoribonuclease) and NSP12 [12,13,14,15]. Structural proteins were encoded by ORF2 to ORF7, including GP2, E, GP3, GP4, GP5, ORF5a, M and N [16]. Recently, NSP2 had been reported as a novel structural component of PRRSV particle, and another two open reading frames, nsp2TF and nsp2N, translated with novel programmed -2/-1 ribosomal frame shifting, were discovered [17].

The PRRSV had a highly restricted cell tropism showing a preference for cells of the monocyte/macrophage in vivo, and PAMs are considered to be the primary target cells of PRRSV. PRRSV can also sustained by MA-104 cell line and its derived cell line, MARC-145 [18,19,20]. Different PRRSV strains show different abilities of infection on PAM, especially between high virulence PRRSV and low virulence PRRSV, and this was also found between HP-PRRSV and its attenuated strain [21]. In the study provided, chimeric viruses were generated by interchanging the 5′UTR + ORF1a, ORF1b, and ORF2-7 + 3′UTR regions between HuN4-F5 and HuN4-F112. ORF1a and ORF2-7 contributed to virus replication [21]. In this study, numbers of chimeric viruses were generated by reciprocally exchanging different regions of PRRSV ORF1a and ORF2-7 between HP-PRRSV HuN4 and HuN4-F112, and the influence of efficiency on PAM cells was tested.

## 2. Materials and Methods

### 2.1. Cells, Viruses, Plasmids and Antibodies

BHK-21 and MARC-145 cells were stored in our lab. Cells were grown at 37 °C in Dulbecco’s modified Eagle’s medium (DMEM, Gibico, Grand Island, NY, USA) supplemented with 10% fetal bovine serum in a humidified 5% CO_2_. Pulmonary alveolar macrophages (PAM) were obtained from 30-day-old piglets that were confirmed to be free of infection with PRRSV, as described previously. The isolated PAMs were cultured in RPMI-1640 medium (Invitrogen, Carlsbad, CA, USA) with 10% heat-inactivated FBS, 100 U/mL penicillin and 10 μg/mL streptomycin in a 37 °C/5% CO_2_ incubator for future studies.

The rescued viruses rHuN4-F5-ORF1a, rHuN4-F5-ORF1b, rHuN4-F5-ORF2-7, rHun4-F112-ORF1a, rHun4-F112-ORF1b, rHun4-F112-ORF2-7, rHuN4-F5 and rHuN4-F112 were used in this study. Two monoclonal antibodies (MAbs) specific to PRRSV M or GP3 protein were prepared and stored by our group, and were used in flow cytometry and Western blotting, respectively.

### 2.2. Construction of Full-Length Chimeric cDNA Clones

Full-length HuN4-F5 and HuN4-F112 infectious clones were utilized as backbones and NSP1-2, NSP2, NSP3-8, ORF2-4 and ORF5-6 were exchanged between rHuN4-F5 and rHuN4-F112, using different strategies (Figure 1). The mutant viruses were first made via site mutation on plasmid NSP1-2 using specific primers (Table 1), and then using the same strategy of p112-NSP1-2H to construct full-length infection clone and rescued viruses.

### 2.3. In Vitro Transcription and Virus Recovery

Virus rescue was performed as previously described. Briefly, all the full-length recombinant cDNA clones were linearized and transcribed to generate capped RNA transcripts using the mMessage High-yield Capped RNA Transcription kit (Ambion, Austin, TX, USA) according to the manufacturer’s instructions. The resulting RNA transcripts were purified with a MEGA Clear kit (Ambion, Austin, TX, USA) and a total of 1 µg RNA was subsequently transfected into BHK-21 cells with DMRIE-C reagent (Invitrogen, Carlsbad, CA, USA). After 24 h, the transfected cells were frozen at −20 °C and thawed. The supernatant samples were collected and passaged on MARC-145 cells. The rescued viruses were confirmed via sequencing, and the 4th passage (on MARC-145 cells) viruses were used in the flowing studies.

### 2.4. Virus Infection Efficiency Test on PAMs

PAMs were initially seeded at a density of 10^6^ cells/well in 6-well plates 4 h prior to infection. These cells were then infected with rHuN4-F5, rHuN4-F112 and other chimeric viruses at a multiplicity of infection (moi) of 2. After 1 h, PAMs were washed with PBS 3 times, cultured in RPMI-1640 medium for another 24 h, and then fixed with 2% formaldehyde for 15 min and permeabilized with 1% saponin for 10 min. The fixed cells were washed 3 times and stained with PRRSV anti m MAb and FITC-conjugated goat anti-mouse IgG for 30 min at 37 °C. Cells were then analyzed using a FACSCalibur cytometer (BD Biosciences, Franklin Lakes, NJ, USA). The infection efficiency was expressed as the percentage of PRRSV positive cells under flow cytometry.

### 2.5. Animal and Experimental Design

Thirty-five 40-day-old PRRSV, PCV2 and CSFV antigen-free and PRRSV antibody-free piglets were selected and randomly divided into nine groups of five piglets each. Piglets in group 1 to 6 were infected with rHuN4-F5, rHuN4-F112, rHuN4-F5-NSP2, rHun4-F112-NSP2, rHuN4-F5-893/979 and rHuN4-F112-893/979 at a dose of 10^5^ TCID_50_ per piglet, intramuscularly. Piglets from group 7 were inoculated with DMEM and served as control. Sera samples were collected at 0, 3, 7 and 14 days post infection (dpi). Rectal temperature was recorded daily and clinical features were observed. Persistent high fever was defined as rectal temperature equal or higher than 40.5 °C and lasting at least 3 days. All animals were euthanized at 14 dpi and necropsied to observe pathological changes in the lungs, which were graded using a lung macroscopic score based on the approximate volume that each lung lobe contributed to the entire lung volume. Briefly, the right cranial lobe, right middle lobe, cranial part of the left cranial lobe and the caudal part of the left cranial lobe each contributed to 10% of the total lung volume, the accessory lobe contributed to 5% and the right and left caudal lobes each contributed 27.5%. Any lung sample which had a macroscopic score equal to or higher than 50 was defined as displaying severe pathological change. Any lung sample which had a macroscopic score higher than 30 and lower than 50 was defined as displaying medium pathological change. Any lung sample which had a macroscopic score equal to or lower than 30 was defined as displaying mild pathological change. Pathological changes in the kidney, spleen, inguinal lymph nodes, tonsil, mesenteric lymph nodes and liver were evaluated and scored from 0 to 3 (from non-pathological to serious pathological) [22]. This study was approved by the Animal Ethics Committee of the School of Harbin Veterinary Research Institute of the Chinese Academy of Agricultural Sciences and was performed in accordance with animal ethics guidelines and approved protocols. The Animal Ethics Committee Approval Number was SYXK (Hei) 2011022.

### 2.6. Viral Copy Detection

The collected and PAMs were used for viral copy detection. For PAMs, 5 × 10^6^ PAM cells were cultured in 6-well plates and infected with rHuN4-F5, rHuN4-F112, rHuN4-F5-NSP2 or rHuN4-F112-NSP2 at a multiplicity of infection (MOI) of 2, followed by incubation at 37 °C for 1 h. Cells were washed and incubated in RPMI-1640 with 2% heat-inactivated FBS, 100 U/mL penicillin and 10 μg/mL streptomycin at 37 °C for another 48 h. The virus-infected supernatants were collected every 12, and viral copies were determined by qRT-PCR. Primers (forward 5′-CCC TAG TGA GCG GCA ATT GT-3′ and reverse 5′-TCC AGC GCC CTG ATT GAA-3′) were designed from the highly conserved ORF 7 region for generation of a 60 bp DNA product. A TaqMan^®^ probe (5′-TCT GTC GTC GAT CCA GA-3′) specific for DNA fragments was labeled with FAM at the 5′ end and MGB at the 3′ end.

### 2.7. Western Blotting

PAMs were infected with rHuN4-F5-NSP2 or rHuN4-F112-NSP2 at a multiplicity of infection (MOI) of 2, then cells were lysed in sample buffer and heated at 95 °C for electrophoresis in 10% SDS-polyacrylamide gels. The separated proteins were transferred to PVDF membranes (Immobilon-P, Millipore, Billerica, MA, USA), blocked for one hour at room temperature in 5% non-fat dry milk in PBS containing 0.05% Tween-20 (Sigma, Saint Louis, MO, USA) (PBST) and incubated with anti-GP3 or anti-β-Actin MAbs (Sigma, Saint Louis, MO, USA) for 1.5 h at room temperature. Then, the membranes were washed 3 times and incubated for 1.5 h at room temperature with 1:2000 diluted HRP-labeled goat anti-mouse IgG antibody (Sigma-Aldrichl, Burlington, MA, USA). Finally, the chemiluminescent ECL Plus substrate (Thermo Fisher Scientific, Crlsbad, CA, USA) was added to the membranes according to the manufacturer’s instructions. Protein bands were visualized with chemiluminescent film (Kodak, Rochester, New York, NY, USA).

### 2.8. Statistical Analysis

Data were presented as averages ± standard deviation (SD). All statistical analyses were performed using SPSS 13.0 statistical software. First, the results of infection efficiency were analyzed among all the groups. This provided the information about changes in each infectious group compared with the control group. Then, the results were divided into two combinations, one combination including chimeric viruses with rHuN4-F5 backbone, the other one including chimeric viruses with rHuN4-F112 backbone. This provided detail about the influence of each changed region based on the same backbone.

## 3. Results

### 3.1. Rescue of Chimeric and Mutant Viruses

All chimeric viruses constructed in the present study were confirmed via DNA sequencing.

### 3.2. NSP2 Primarily Contributes to the Infection Efficiency in PAMs

Significant lower infection efficiency of rHuN4-F112 was found on PAMs compared with rHuN4-F5. When ORF1a was exchanged, the infection efficiency of rHuN4-F5-ORF1a decreased dramatically compared with rHuN4-F5, and significantly increased infection efficiency of rHuN4-F112-ORF1a was found compared with rHuN4-F112. Slight changes in PRRSV positive rates were observed on PAMs infected with chimeric rHuN4-F112-ORF1b, rHuN4-F112-ORF2-7, rHuN4-F5-ORF1b and rHuN4-F5-ORF2-7 compared with parent viruses rHuN4-F112 or rHuN4-F5 (Figure 2A,D).

For future determination of the regions which contribute to the infection efficiency on PAMs, NSP1-2, NSP3-8, ORF2-4 and ORF5-6 were exchanged between rHuN4-F5 and rHuN4-F112. Results showed that significant differences were found when the parent viruses changed the NSP2 region; however, changes were also found between rHuN4-F112-ORF2-4 and rHuN4-F112 (Figure 2B,D).

### 3.3. Determination of the Key Amino Acids in NSP2 Gene

There were total of eight amino acid substitutions in NSP2 gene of the attenuated vaccine strain HuN4-F112, as aligned with that of the parental strain HuN4. Site-directed mutations in amino acids at 401, 477, 799, 1060 and 1136 did not significantly change viral infection efficiency on PAMs; however, mutations of amino acids at 893 and 979 dramatically changed the efficiency of rHuN4-F5 and rHuN4-F112 (Figure 2C,D).

### 3.4. NSP2 Influence Virus Replication In Vitro

At 36 h post-infection, viral copies of rHuN4-F112, rHuN4-F112-NSP2 and rHuN4-F5-NSP2 were significantly lower compared with rHuN4-F5. No significant difference was found between rHuN4-F112, and rHuN4-F112-NSP2 (Figure 3A). Results of Western blot showed that when NSP2 was exchanged, the expression of GP3 was increased dramatically in the rHuN4-F112-NSP2 group compared with rHuN4-F112; however, significantly lower GP3 expression was found in rHuN4-F5-NSP2 compared with rHuN4-F5 (Figure 3B).

### 3.5. Clinical Performance after Infection

After infection, piglets infected with rHuN4-F5 manifested various diseases, including persistent high fever and anorexia, and four piglets died. Persistent high fever was found in piglets belonging to the rHuN4-F5 group from 4 to 6 dpi; however, piglets in rHuN4-F112 did not show any clinical symptoms. Two and one piglets died in the rHuN4-F5-NSP2 and rHuN4-F112-NSP2 groups, respectively. None of the piglets in these groups showed persistent high fever. Four piglets died in the rHuN4-F5-893/979 group and persistent high fever was found from 4 to 9 dpi. No clinical symptoms were found in the rHuN4-F112-893/979 groups (Figure 4A,B).

### 3.6. Tissue Necropsy Changes after Infection

The macroscopic scores showed that lungs infected with rHuN4-F5 exhibited serious pathological changes. A lower lung score was found in the rHuN4-F5-NSP2 group, although there was no significant difference. Dramatic lower lung score was found in the rHuN4-F112 group compared with the rHuN4-F112-NSP2 and rHuN4-F112-893/973 groups. More severe and medium pathological cases were also found in the rHuN4-F112-NSP2 and rHuN4-F112-893/973 groups (Table 2).

Piglets in rHuN4-F5 group showed more severe tissue changes than the other groups. Compared with rHuN4-F5 group, lower tissue scores for kidney, spleen, liver and mesenteric lymph nodes were found in rHuN4-F5-NSP2 and rHuN4-F5-893/973; higher tissue scores of kidney, spleen, liver and mesenteric lymph nodes were found in rHuN4-F112-NSP2 compared with rHuN4-F112, although no significant difference was found (Table 3).

### 3.7. Viremia Copies in Sera

Significant higher virus copies were found in group rHuN4-F5 compared with other groups at 3 (with the exception of rHuN4-F5-NSP2 group) and 7 dpi (with the exception of the rHuN4-F5-893/979 group) (Figure 5A). For the chimeric viruses with rHuN4-F5 backbone, significant lower viremia was found in the rHuN4-F5-NSP2 and rHuN4-F5-893/979 groups at 3 and 7 dpi (Figure 5B). For the chimeric viruses with rHuN4-F112 backbone, significant higher viremia was found in the rHuN4-F112-NSP2 group at 3 dpi (Figure 5C).

## 4. Discussion

Previous observations indicated that the attenuated live vaccine strain of PRRSV did not establish sufficient infection in PAMs compared with the wild-type strain both in vivo and in vitro. Therefore, the ability of live PRRS vaccine viruses to replicate in PAMs was considered to be a key feature in distinguishing them from wild PRRSV isolates. The wild-type PRRSV replicates preferentially in differentiated PAMs, which explained why PAMs were the first to be used successfully in PRRSV isolation and as a good model for PRRSV respiratory challenge.

In the past few years, increasingly accumulated data have demonstrated that PRRSV NSP2 is a multifunctional protein involved in viral replication and antiviral innate immune responses [23,24,25,26,27]. The relationship of PRRSV NSP2 and virus attenuation has been described briefly in previous studies. Kim et al. constructed recombinant PRRSV with deletion of 131 amino acid deletion within a relatively conserved region of NSP2 and found that this virus grew normally in MARC-145 cells and PAMs, but was less virulent in pigs, as examined via gross and micro-histopathology [28,29]. Wang et al. examined NSP2 expression of HP-PRRSV TJ and attenuated TJM in MARC-145 cells and concluded that the downregulation of NSP2 significantly decreased PRRSV replication and expression of structural proteins in MARC-145 cells, but the viral replication in PAMs was not studied [30]. In the present study, we found that ORF1a was important for the infection efficiency of PRRSV, and then NSP2 was demonstrated to be critical and influence virus replication on PAMs. Similar results were provided by other studies [23,25,31,32,33,34,35,36]. Furthermore, 893aa and 979aa were demonstrated to play a key role in the infection efficiency during PRRSV infection of PAMs. The non-structural protein 2 may influence virus infection due to its structural protein formation, which was reported in [37,38]. In addition to PRRSV NSP2, other genes may also be involved in viral replication of PRRSV in PAMs, as a slight increase in viral replication was observed in PAMs infected with the chimeric virus rHuN4-F112-ORF2-7.

The replication changes always indicated the virulence change. We further test the influence of NSP2 and amino acid 893+979. The survival rate clearly showed that NSP2 was important for virulence, and the results of tissue damage also proved this. As one of the most important characteristics of HP-PRRSV infection, persistent high fever disappeared in the rHuN4-F5-NSP2 group, and lower viremia was also found compared with rHuN4-F5. These results provide more evidence that NSP2 influences virulence. These results were consistent with those of other studies reporting that 3′UTR, Nsp1-2, Nsp3-8 and Nsp10-12 were virulence determinants [21,33,34]. Interestingly, when NSP2 was changed, rHuN4-F112-NSP2 did not show the same dramatic results as rHuN-F5-NSP2; however, different regions had different susceptibility to wild-type virus and vaccine strain, which was also observed in our previous study. This indicated that different mechanisms might be used during PRRSV virulence enhancement in vivo and attenuation in vitro [21].

In conclusion, we demonstrated that NSP2 contributed to the infection efficiency of PRRSV; 893aa and 979aa made the difference between HuN4-F5 and HuN4-F112. Then, an in vivo study demonstrated that NSP2 were not only important for infection efficiency in vitro, but also influenced the virulence. This was indicated by the results of survival rate, temperature, viremia, lung score and tissue score. However, although 893aa and 979aa were important in the in vitro study, they did not play a key role in the virulence change. These results will facilitate better understanding of the full functions of NSP2 in virology, vaccinology and pathogenesis of PRRSV in future study, and they will provide a deep understanding of virus attenuation.

## Figures and Tables

**Figure 1 viruses-14-02613-f001:**
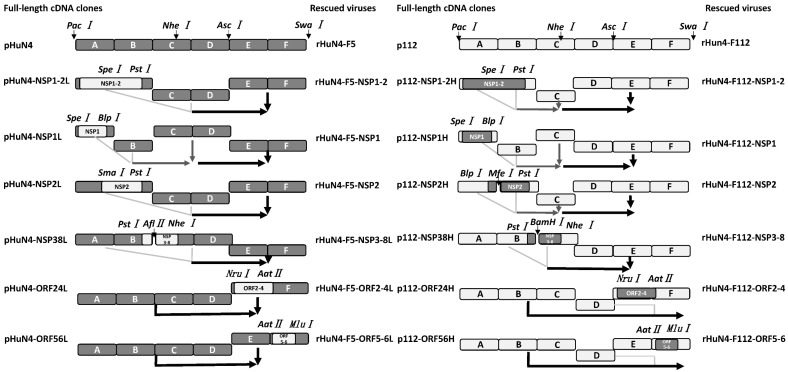
Construction of chimeric PRRS viruses. The PRRSV genome structure of rHuN4-F5 (dark gray) and rHuN4-F112 (light gray) were shown at the top of the figure. Restriction sites used for cloning were set in italics. The full infection clone plasmids were shown on the **left** and the designation of each chimeric viruses were shown on the **right**.

**Figure 2 viruses-14-02613-f002:**
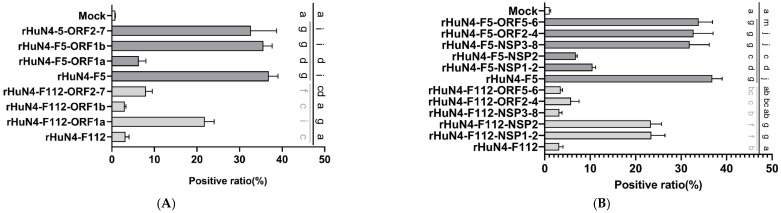
Infection efficiency of the chimeric viruses. (**A**): Infection efficiency of the chimeric viruses with ORF1a, ORF1b and ORF2-7 exchanged. (**B**): Infection efficiency of the chimeric viruses with NSP1-2, NSP2, NSP3-8, ORF2-4 and ORF5-6 exchanged. (**C**): Infection efficiency of the chimeric viruses with site mutations. (**D**): Flow cytometric analysis of infection efficiency. Chimeric viruses with rHuN4 backbone were showed with dark gray. Chimeric viruses with rHuN4-F112 backbone were showed with light gray. Results were showed with single alphabet, or a range of alphabets which were represented by the first and last alphabet. The results which share same alphabet mean *p* ≥ 0.05; results that do not share same alphabet but have adjacent alphabets with each other mean *p* < 0.05; results that do not share same alphabet or adjacent alphabets with each other mean *p* < 0.01. Results among all the viruses were shown with black alphabets above the line; results among rHuN4-F5 backbone chimeric viruses were shown with black alphabets below the black line, and results among rHuN4-F112 backbone chimeric viruses were shown with light gray alphabets below the gray line.

**Figure 3 viruses-14-02613-f003:**
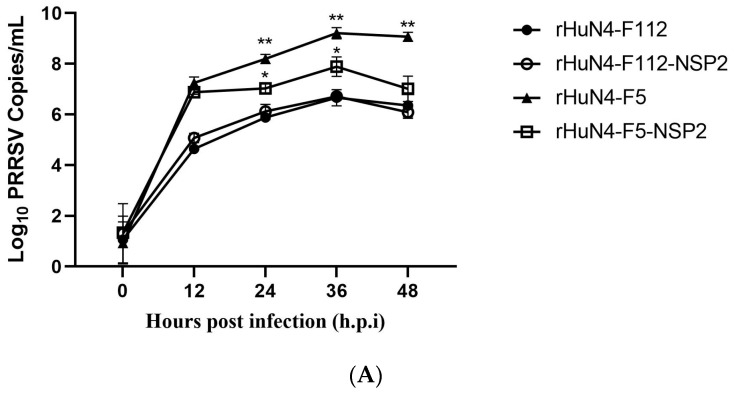
Virus replication on PAMs. (**A**): Viral copies of the chimeric viruses. (**B**): GP3 protein expression after infection. * *p* < 0.5, ** *p* < 0.01.

**Figure 4 viruses-14-02613-f004:**
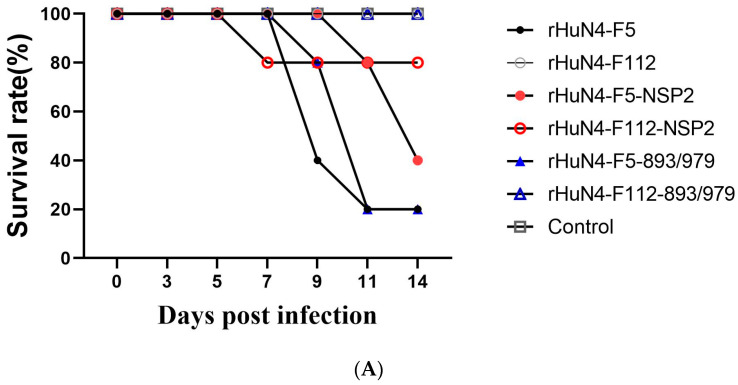
Survival rate and rectal temperature after infection. (**A**): Survival rate of each group. (**B**): Mean rectal temperature of each group after infection. Mean rectal temperatures were presented as means ± standard deviations (error bars).

**Figure 5 viruses-14-02613-f005:**
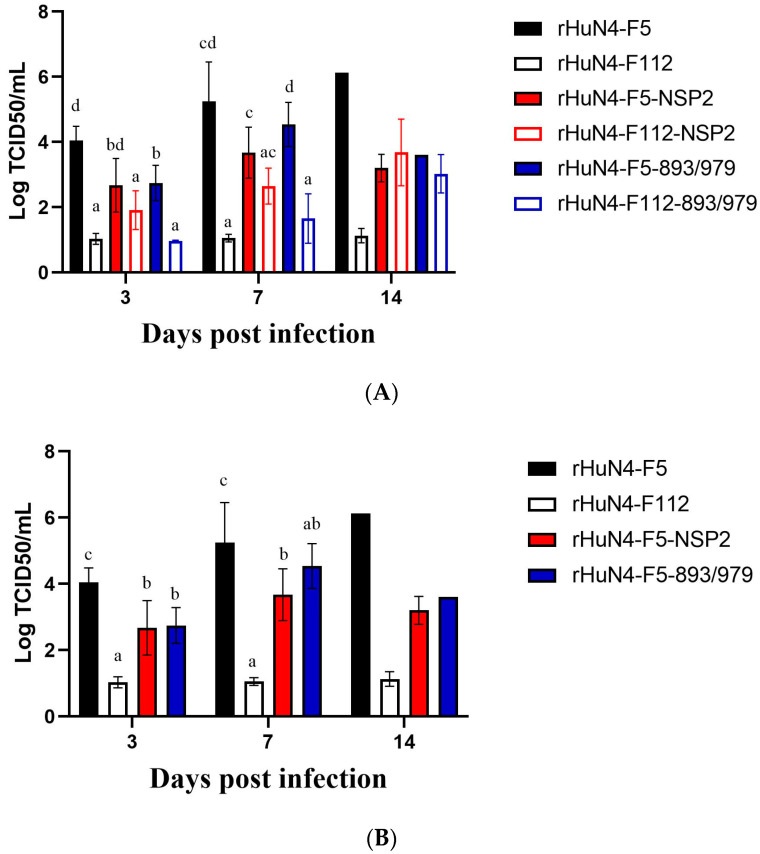
Viremia of each group after infection. The viremia of the chimeric viruses and their parental virus was shown with means ± standard deviations. Results were shown with single alphabet, or a range of alphabets which were represented by the first and last alphabet. The results which share same alphabet mean *p* ≥ 0.05; results which do not share same alphabet, but have adjacent alphabets with each other mean *p* < 0.05; results which do not share same alphabet or adjacent alphabets with each other mean *p* < 0.01.

**Table 1 viruses-14-02613-t001:** Information of primers and chimeric viruses.

Primers	Sequences (5′-3′)	Information	Chimeric Viruses
Q-RT401F Q-RT401L	CTATGGTCGCTCGTCACGCTTC TGGACGAAGCGTGACGAGCGAC	Primers used for mutated NSP2 401aa from rHuN4-F5 to rHuN4-F112	rHuN4-F5-401/477
Q-RT477F Q-RT477L	GATCTTGTGAACATCATCCAAA CTGAGGATTTGGATGATGTTCA	Primers used for mutated NSP2 477aa from rHuN4-F5 to rHuN4-F112	
Q-RT799F Q-RT799L	GGCTGAGCAGGTCAATTTGAAAGCT ACCCAAGCTTTCAAATTGACCTGCT	Primers used for mutated NSP2 799aa from rHuN4-F5to rHuN4-F112	rHuN4-F5-799
Q-RT893F Q-RT893L	TCGCGACGTGCCCCCAAGCTGATGA AAGGTGTCATCAGCTTGGGGGCACG	Primers used for mutated NSP2 893aa from rHuN4-F5 to rHuN4-F112	rHuN4-F5-893 and rHuN4-F5-893/979
Q-RT979F Q-RT979L	CTGTGTCATCAAGCAGCCCCTTGTC CTTAATACTTGACAAGGGGCTGCTT	Primers used for mutated NSP2 979aa from rHuN4-F5 to rHuN4-F112	rHuN4-F5-979 and rHuN4-F5-893/979
Q-RT1060F Q-RT1060L	CGTTTCGCATCTTAAGTGGCAGGTT AAACTCAAACCTGCCACTTAAGATG	Primers used for mutated NSP2 1060aa from rHuN4-F5 to rHuN4-F112	rHuN4-F5-1060/1136
Q-RT1136F Q-RT1136L	CTCTAAGGGAGAACCGGTCAGTGAC GGCAGGTTGGTCACTGACCGGTTCT	Primers used for mutated NSP2 1136aa from rHuN4-F5 to rHuN4-F112	
R-QN401U R-QN401L	CTATGGTCGCTCATCACGCTTC AGCGGACGAAGCGTGATGAGCG	Primers used for mutated NSP2 401aa from rHuN4-F112 to rHuN4-F5	rHuN4-F112-401/477
R-QN477F R-QN477L	GATCTTGTGAACACCATCCAAA GAGGATTTGGATGGTGTTCACA	Primers used for mutated NSP2 477aa from rHuN4-F112 to rHuN4-F5	
R-QN799F R-QN799L	TGGGCGGCTGAGCAGGTCGATTTAA CCAAGCTTTTAAATCGACCTGCTCA	Primers used for mutated NSP2 799aa from rHuN4-F112 to rHuN4-F5	rHuN4-F112-799
R-QN893F R-QN893L	GTCGCGACGTGTCCCCAAGCTGATG GTCATCAGCTTGGGGACACGTCGCG	Primers used for mutated NSP2 893aa from rHuN4-F112 to rHuN4-F5	rHuN4-F112-893 and rHuN4-F112-893/979
R-QN979F R-QN979L	GTGTCATCAAGCAGCCCCTTGTCAA ATCTTAATACTTGACAAGGGGCTGC	Primers used for mutated NSP2 979aa from rHuN4-F112 to rHuN4-F5	rHuN4-F112-979 and rHuN4-F112-893/979
R-QN1060F R-QN1060L	GTTTCGCATCTTAAATGGCAGGTTT AACTCAAACCTGCCATTTAAGATGC	Primers used for mutated NSP2 1060aa from rHuN4-F112 to rHuN4-F5	rHuN4-F112-1060/1136
R-QN1136F R-QN1136L	CAAGGGAGAACCGGTCTGTGACCAA TTGGCAGGTTGGTCACAGACCGGTT	Primers used for mutated NSP2 1136aa from rHuN4-F112 to rHuN4-F5	

**Table 2 viruses-14-02613-t002:** Mean values (±SD) of macroscopic scores of lung tissues after infection.

Designation	Number	Macroscopic (Lung)			
		Mean ± SD	Pathological Changes (Score)		
			≤30	30 to 50	≥50
rHuN4-F5	5	89 ± 9.64 ^d^	0	0	5
rHuN4-F5-NSP2	5	77.4 ± 31.94 ^d^	1	0	4
rHuN4-F5-893/979	5	90.5 ± 3.55 ^d^	0	0	5
rHuN4-F112	5	23.2 ± 15.42 ^a^	5	0	0
rHuN4-F112-NSP2	5	26.8 ± 25.47 ^b^	2	2	1
rHuN4-F112-893/979	5	53.8 ± 21.87 ^b,c^	1	2	2
Control	5	17.6 ± 17.67 ^a^	4	1	0

Superscript letters which share same alphabet mean *p* ≥ 0.05; those which do not share same alphabet, but have adjacent alphabets with each other mean *p* < 0.05; those which do not share same alphabet or adjacent alphabets with each other mean *p* < 0.01.

**Table 3 viruses-14-02613-t003:** Pathological examination of piglets infected with chimeric viruses.

Designation	Macroscopic (Organs)					
	Kidney	Spleen	Tonsil	Liver	Mesenteric Lymph Nodes	Inguinal Lymph Nodes
rHuN4-F5	3 ± 0 ^c^	1.4 ± 1.14	1 ± 1	2 ± 0.7 ^b^	2.2 ± 1.1	0.4 ± 0.55
rHuN4-F112	0 ± 0 ^a,c^	0.6 ± 1.34	0 ± 0	0 ± 0 ^a^	0 ± 0	0.8 ± 1.09
rHuN4-F5-NSP2	1.2 ± 1.1 ^a,c^	0.2 ± 0.44	1.4 ± 1.34	1.4 ± 0.89 ^a,b^	0.6 ± 0.89	1.8 ± 1.3
rHuN4-F112-NSP2	0.4 ± 0.89 ^a,b^	1 ± 1.4	0 ± 0	0.8 ± 1.09 ^a,b^	1 ± 1.44	1.2 ± 1.64
rHuN4-F5-893/979	1.4 ± 1.14 ^a,c^	0.4 ± 0.55	1.6 ± 1.14	1.2 ± 0.83 ^a,b^	1.4 ± 1.34	1 ± 0.7
rHuN4-F112-893/979	0.2 ± 0.45 ^a^	0 ± 0	0.4 ± 0.89	0 ± 0 ^a^	0.2 ± 0.45	0.4 ± 0.55
Control	0.4 ± 0.55 ^a^	0 ± 0	0 ± 0	0 ± 0 ^a^	0 ± 0	0.8 ± 1.1

For the same organ, superscript letters which share same alphabet mean *p* ≥ 0.05; those which do not share same alphabet but have adjacent alphabets with each other mean *p* < 0.05; those which do not share same alphabet or adjacent alphabets with each other mean *p* < 0.01. Non-superscript letters mean no significant difference.

## Data Availability

Not applicable.

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
