# Peer review of "Nonstructural Protein 2 Is Critical to Infection Efficiency of Highly Pathogenic Porcine Reproductive and Respiratory Syndrome Virus on PAMs and Influence Virulence In Vivo"

_viruses, 2022, doi:10.3390/v14122613_

Round 1

Reviewer 1 Report

In this manuscript, the authors provided some new insight into the nsp2 function of PRRSV, which will contribute to a better understanding of the PRRSV cellular tropism and virulence change. However, there are still some concerns that need to be addressed before the acceptance.

1. Although the evaluation criterion of tissue necropsy changes was defined from 0 to 3,  the particular state should be described.   

2. The nsp2 can influence the viral replication of PRRSV in  PAMs, it is wondered if the lower infection efficiency of rHuN4-F5-NSP2(compared with rHuN4-F5) was due to the lower capability of absorption, release, or packaging.

Author Response

1.Although the evaluation criterion of tissue necropsy changes was defined from 0 to 3,  the particular state should be described.   

A: The reference(22) was add.

2.The nsp2 can influence the viral replication of PRRSV in  PAMs, it is wondered if the lower infection efficiency of rHuN4-F5-NSP2(compared with rHuN4-F5) was due to the lower capability of absorption, release, or packaging

A: It’s reasonable to know the mechanism of how NSP2 influence infection efficiency not only from virus itself, but also from the virus-host interaction. There was no influence on absorption(data not shown), however the influence on release and packaging were not tested and should be studied in future.

Reviewer 2 Report

1, line 130, there is a problem with the grouping. It should be divided into seven groups, right? Also Piglets in group 1 to 4 were infected with should be 1 to 6. Please revise the corresponding part well.
2, line 135, the serum sample collection time why choose 0, 3, 7, 14 days, please do explain and attach references.
3. It is suggested to add the experimental data of virus detoxification to provide a more favorable proof for the consistency of virus proliferation in vitro and in vivo.

Author Response

1, line 130, there is a problem with the grouping. It should be divided into seven groups, right? Also Piglets in group 1 to 4 were infected with should be 1 to 6. Please revise the corresponding part well.

A: The mistake was corrected and corresponding part was re-checked.

2, line 135, the serum sample collection time why choose 0, 3, 7, 14 days, please do explain and attach references.

A: Normally, the acute infection period of PRRSV infection was from 1 to 14 dpi, then the clinical response might be caused by PRRSV or by co-infection with other pathogens. The peak of viremia were always 3 or 7 dpi, and occasionally at 14dpi, so 1,3,7 and 14dpi were chose.

  1. It is suggested to add the experimental data of virus detoxification to provide a more favorable proof for the consistency of virus proliferation in vitro and in vivo.

A: It’s a good suggestion, however due to the limitation of Animal testing facility, we do not use virus detoxification, it should be considered in the future study.